# Method for Identification of Aberrations in Operational Data of Maritime Vessels and Sources Investigation

**DOI:** 10.3390/s24072146

**Published:** 2024-03-27

**Authors:** Jie Cai, Marie Lützen, Adeline Crystal John, Jakob Buus Petersen, Niels Gorm Maly Rytter

**Affiliations:** 1Department of Technology and Innovation, University of Southern Denmark, Campusvej 55, 5230 Odense, Denmark; ngry@iti.sdu.dk; 2Department of Mechanical and Electrical Engineering, University of Southern Denmark, Campusvej 55, 5230 Odense, Denmark; mlut@sdu.dk; 3Vessel Performance Solution ApS, Diplomvej 381, 2800 Kgs Lyngby, Denmark; acj@vpsolutions.dk (A.C.J.); jbp@vpsolutions.dk (J.B.P.)

**Keywords:** ship operations, data analysis, noon reports, autolog data

## Abstract

Sensing data from vessel operations are of great importance in reflecting operational performance and facilitating proper decision-making. In this paper, statistical analyses of vessel operational data are first conducted to compare manual noon reports and autolog data from sensors. Then, new indicators to identify data aberrations are proposed, which are the errors between the reported values from operational data and the expected values of different parameters based on baseline models and relevant sailing conditions. A method to detect aberrations based on the new indicators in terms of the reported power is then investigated, as there are two independent measured power values. In this method, a sliding window that moves forward along time is implemented, and the coefficient of variation (CV) is calculated for comparison. Case studies are carried out to detect aberrations in autolog and noon data from a commercial vessel using the new indicator. An analysis to explore the source of the deviation is also conducted, aiming to find the most reliable value in operations. The method is shown to be effective for practical use in detecting aberrations, having been initially tested on both autolog and noon report from four different commercial vessels in 14 vessel years. Approximately one triggered period per vessel per year with a conclusive deviation source is diagnosed by the proposed method. The investigation of this research will facilitate a better evaluation of operational performance, which is beneficial to both the vessel operators and crew.

## 1. Introduction

The operational data that come from vessels are one of the most important sources with practical usage. Among others, in the pre-fixture stage of voyage planning, these data facilitate shipping companies finding the most navigated routes [1], which optimizes the sailing distance calculation and budget estimation. In addition, these data are normally adopted to monitor and evaluate vessel sailing performance by shipping companies [2].

In the endeavors of the shipping industry to meet the Paris agreement [3] for the improvement of its environmental footprint, vessel performance has become an essential issue that needs to be further optimized. Generally, the differences between practical vessel performance and a predefined baseline (defining ideal ship performance) should be minimized. Therefore, such operational data should be reliable enough for a proper ship performance estimation. However, they are normally subject to many sources of uncertainty, including faulty sensors, manual data collection, and even deliberate data manipulation, resulting in large deviations for the estimation of true ship performance. In this work, the period of recorded data variation that deviates from expected variations is defined as the aberration. To discover proper insights from such error-prone data and reap benefits, it is necessary to develop practical methods to detect aberrations and work towards automatic sources and, eventually, root cause identification.

In shipping, there exist various ways to collect operational data from vessels. One is the noon report, which is filled out daily by the vessel crew. With the advancement of sensing and sensor technologies, autolog systems are utilized to automatically collect data onboard from various sensors [4,5,6]. Such operational data can be used to better understand and optimize navigation, energy consumption, emissions, and maintenance [7,8]. With the rapid development of new technology, including the Internet of Things (IoT), real-time connectivity, and artificial intelligence (AI) [9,10], looking at operational data and performance monitoring brings focus on quality diagnostics and root cause analysis, with the aim of the sustainable operation of shipping in the near future. The direct utilization of operational data is still limited due to the lack of standardized and reliable data sources. The application and implementation of machine learning (ML) is challenging due to the above-mentioned issues and other data quality problems. Hence, there is a necessity to further diagnose these phenomena.

Recent years have witnessed a considerable amount of research on vessel operational data. Cai et al. [1] developed voyage routes from automatic identification system (AIS) data based on clustering methods. Cai and Lützen [11] applied a sliding window to detect pattern changes in specific fuel oil consumption (SFOC) in noon reports. Dalheim and Steen [12] developed a method to monitor the change-point of vessel operational data from sensors based on hypothesis testing of samples inside a moving window, assuming that the underlying samples can be modeled by a deterministic linear trend model. The same authors in [13] proposed a stepwise framework to deal with the possible quality problems that occur in sensors and device data from vessel operations. Similar research and utilization of vessel operational data can also be found in Shelmerdine [14], Le Tixerant et al. [15], and Han and Yang [16].

However, there is still a lack of research on the identification of aberrations, as defined above, and especially a lack of effective methods to reveal the corresponding underlying reasons. These aberrations should be flagged, and the sources should be identified automatically in order to facilitate vessel operations. Therefore, the objective of this research is to investigate a practical method to identify aberrations in vessel operation based on both manual and sensor data that is capable of triggering alerts and, subsequently, concluding the underlying sources of such aberrations.

This paper is organized as follows: In Section 2, operational data are described, and the data of two vessels from the container shipping company Hapag-Lloyd [17] are compared. In Section 3, the methodology to identify data aberrations in vessel operations is elaborated upon, including baseline models and traditional indicators for ship performance, new proposed indicators, and the sliding window method. Section 4 contains the results of a case study applying the proposed method to the new indicator using data from a commercial vessel. In Section 5, the discussion and method limitations are further presented. Finally, the conclusions are drawn in Section 6.

## 2. Data Sources: Noon Reports and Autolog Data

In this section, the two types of data sources in vessel operations, noon reports and autolog data, are first described. Both of them will be used for the identification of data aberrations in Section 3. The statistical characteristics of the two data sources recording the same operational process are compared based on two vessels from the container operator Hapag-Lloyd. The results provide a glimpse of data source differences in practice. However, generic results cannot be derived due to the limited number of cases. The effect of using different data sources on the accuracy of the method is not studied in this paper.

Operational information from a vessel is recorded in different ways for regulatory and monitoring purposes. The noon report is one of the classical ways to record data. A noon report summarizes relevant sailing information and is manually prepared by vessel crew approximately every 24 h and shared with onshore offices. The noon report includes information such as sailing time, sailed distance, course, engine run time, power produced, total fuel consumption, and weather conditions. These values are used in various performance analyses, for example, to monitor the degradation of vessel performance or to measure emissions.

Another method of collecting vessel operational data is using autolog systems that record information automatically through in-service monitoring systems and sensors deployed onboard. Autolog data from sensors and equipment onboard have, over the past few years, become more popular in the area of digital ship operations [2]. During a voyage, the operational data can be constantly logged at a high frequency, thereby capturing instantaneous variations in different parameters. The autolog system has been gradually improved and embraced by industry with the development of IoT technology, and it is expected to provide more reliable and accurate data than noon reports. The sample frequency of autolog data is flexible and normally set to be in minutes. However, the data are prone to be erroneous due to factors such as malfunctioning, unexpected sensor shifting, and asynchronization.

For comparison of the two data sources, the traditional performance indicator SFOC (see Section 3.1.1) is used. Note that the frequency of the autolog dataset has been down-sampled to hours (between 1 and 2 h) for better comparison, while the frequency of the noon report is roughly 1 day. Figure 1 and Figure 2 show the results of the SFOC variations from noon and autolog data for vessel A and vessel B, respectively. The SFOC values in each sample are normalized by the SFOC baseline from shop tests performed by the engine manufacturers. Figure 3 and Figure 4 illustrate the statistical distribution of the different operational data sources. Results show that, for vessel A, the noon report has a lower median value (from 99.2% to 105.2%) and a similar standard deviation (from 7.41% to 7.06%) compared to the autolog data. A simple hypothesis test is conducted as well, which is to test if the two mean values (here we use the mean value of the SFOC percentage: 100.83% and 106.0% for the noon report and autolog, respectively) are statistically equal with the null hypothesis as $H_{0}:\mu =\mu_{0}$, where μ is the mean value of the SFOC from the noon report and $\mu_{0}$ is the expected mean value of the SFOC from the autolog data. Since the population standard deviation is unknown, the sample standard deviation value is used, which is 7.41% in this case. The number of recorded data is 734. Thus, the *t* value is calculated as −18.9. If the level of significance (α) is selected as 0.05, the significant value ($t_{\alpha /2}$) can be then estimated from the student’s t-distribution table, which is 1.963. Obviously, such an assumption is rejected, which means that the mean values of the two recorded datasets from vessel A cannot be considered the same statistically.

For vessel B, the noon report has a lower standard deviation (from 8.2% to 9.7%) and a nearly identical median value (from 98% to 98.3%) compared to the autolog data. More unexpected spikes are observed in the noon reports than in the autolog data for both vessels. The statistical distribution also shows that autolog data have a larger spread between the sample minimum and sample maximum but with fewer outliers, as indicated by the much longer whiskers in the left diagrams of both Figure 3 and Figure 4. The same hypothesis test conducted for vessel A is conducted for vessel B to test if the mean values of the SFOC percentage are the same. Surprisingly, the calculated *t* value is 0.26, and the null hypothesis can be accepted, which means that the mean values recorded from the two datasets can be considered the same statistically.

It can be concluded that the two data sources represent similar vessel operational behavior but with statistical discrepancies due to different data acquisition methods and many uncertainties, such as sensor derivations under extreme environments and human errors. In the case of noon reports, e.g., vessel A, human factors may be much larger, causing more outliers, higher spikes, and statistical differences between mean values. The smaller standard deviation indicates that noon reports cannot sufficiently represent operational details due to the 24 h time interval. The autolog data have higher variation, so they are capable of capturing realistic operational details with fewer outliers and fewer spikes but introduce more noise. It should be noted that the frequency effect has not been investigated, which may introduce other discrepancies.

## 3. Methodology

In this section, the practical method for data aberrations is proposed. The model can be applied to operational datasets from either noon reports or autolog data. Note that, in the following work, if there is no explicit indication, both types of operation data are applicable. In order to better introduce this method, the baselines for performance analysis and relevant traditional indicators in operations are first introduced, as seen in Section 3.1. Then, new indicators for data aberrations are developed in Section 3.2, which are the parameter errors calculated by our Trinity model (see Section 3.1.2). Using these new indicators, the sliding windows are applied to identify data aberrations, as illustrated in Section 3.3.

### 3.1. Performance Baselines and Traditional Indicators

When analyzing performance from vessels, traditional indicators, such as RPM, power, speed, SFOC, and resistance, are usually adopted, and the baseline models for them are vital to mutually defining ideal performance in operation, given different factors such as sea states, draft, and fouling levels. These models are normally determined by physical laws and binding data such as shop tests from the engine manufacturer and sea trials delivered by the shipyard. The baseline models used in this analysis and the relevant indicators are elaborated below.

#### 3.1.1. Engine Baselines

The indicator SFOC is the measure of the mass of fuel consumed by the engine per unit time to produce a unit of power. It is unit-engine-specific, and normally given as the function of the engine load in the percentage of maximum continuous rating (MCR). Figure 5 shows an example of the main engine SFOC of vessel C’s model from the tanker shipping company TORM [18]. The main engine SFOC model is an important baseline that can be used to estimate excess fuel consumption and evaluate engine power based on the fuel consumed in operations.

Figure 6 shows another baseline model, which is the main engine load diagram, presenting the RPM–power limits of the engine of vessel A. BL_continuous denotes the continuous running condition of the engine when ample air is present in the combustion chamber to secure acceptable combustion and the maximum allowable loads are not exceeded. The BL_overload represents the overload operation limit of the engine, which is possible only for limited periods (e.g., 1 h per 12 h or when required in an emergency situation) [19]. The BL_heavy_propeller baseline represents the ideal power for continuous operation. The light propeller curve BL_light_propeller is the relation of power and RPM for the propeller with a clean hull and in calm weather. This curve shifts towards the BL_heavy_propeller because of heavy weather and a fouled hull, and the distance between these two curves is given by the light running margin (LRM). Details regarding the LRM are elaborated upon in Section 3.1.2.

The RPM–power baseline can be used to estimate vessel performance with the recorded power values in operations. However, in practice, there are two different methods to measure vessel power: one is the calculated value based on the measured fuel consumption from the fuel flow meter and the engine SFOC (the red dots in Figure 6), and the other is the direct measurement from torsion meters deployed on the shaft line (to measure the torque) and the RPM (the green dots in Figure 6). The problem is which one people should choose for a better estimation. Ideally, the powers obtained from these two methods should be the same since they reflect the same vessel operational process. However, there exist deviations in reality, as described previously, resulting in different estimates of the vessel performance. Hence, one source must be a more consistently reliable measurement than the other, and the choice of the right source will further affect the evaluation accuracy of operational performance.

#### 3.1.2. Speed–Power–RPM Baselines

According to the physical laws and specific designs of vessels, there is a fixed relation between speed, power, and RPM, which will form the basis of the new proposed indicators (seen in Section 3.2) in this work. This speed–power–RPM baseline of the vessel is, hereafter, referred to as the Trinity model. Note that the Trinity model is vessel design-specific and is determined based on several factors, including the principal dimensions of the vessel, the design speed, power, and RPM, and, most importantly, the vessel’s own sea trial performance. In addition, the baseline models are only for vessels with a fixed-pitch propeller. These curves form the basis of the calculation of expected relations (speed–power–RPM) accounting for specific loading conditions, draft, and weather during operations.

The process of developing this model is briefly explained here. The Trinity model estimate starts with the evaluation of the power (*P*) required by vessels to maintain the sailing speed (*V*) given the resistance ($R_{T}$) of the vessel, as seen in Equation (Equation 1).
(1)$$P=R_{T}\cdot V$$

In practice, the vessel resistance is normally estimated by semi-empirical methods that are widely used in the literature since they are relatively simple and reasonably accurate [20,21,22]. Equation (Equation 2) expresses the semi-empirical equation of vessel resistance, where *S* is the hull-wetted surface area at the draft, ρ is the density of sea water, *V* is the measured speed, and $C_{T}$ is the total resistance coefficient, which can be influenced by factors such as wind, wave, current, types of propeller, water depth, and hull biofouling. Therefore, if the estimated resistance is provided, the expected power can be estimated (Equation (Equation 1)).
(2)$$R_{T}=\frac{1}{2}\cdot C_{T}\cdot \rho \cdot S\cdot V^{2}$$

To build the relation between power and RPM at the design point, the LRM is used, which can be expressed as Equation (Equation 3) [19], where $RPM_{light}$ and $RPM_{heavy}$ are the propeller RPM at the light and heavy load conditions, respectively. The relation of the power and RPM is constructed by the relation $P\propto {\left(LRM\cdot RPM\right)}^{3}$ according to the so-called “cubic law” based on hydrodynamic principles [23].
(3)$$LRM=\frac{RPM_{light}-RPM_{heavy}}{RPM_{heavy}}\cdot 100$$

In this fashion, the speed–power–RPM Trinity model of the newly built vessel in calm sea conditions is constructed as a baseline. This model can easily be calibrated to different weather conditions and degrees of hull and propeller fouling. Due to confidentiality requirements from the shipping company, we cannot provide further information on specific resistance coefficients, hull degradation due to fouling, or other relevant factors. Therefore, the “expected” value of the different parameters can be estimated based on the Trinity model with different input subsets in vessel operations. Note that, in this analysis, the vessel sea trial performance is used as the baseline of evaluating hull degradation over time. Figure 7 shows an example of the speed–power and speed–RPM plots for vessel A at the draft of 14.5 m.

### 3.2. New Indicators for Data Aberrations

#### 3.2.1. Motivation

As previously described in Section 3.1.1, there exist two measured power values: one based on the fuel consumption and the other based on the torsion meter. The proper choice of recorded power will further affect the accuracy of performance estimation. Meanwhile, operational data are subject to many sources of uncertainty, such as faulty sensors and manual data collection, introducing even more challenges to accurately estimate ship performance from the data-driven standpoint. Therefore, one big motivation is to propose new indicators to counteract such uncertainties, which will, in turn, facilitate the selection of the more accurate parameters for performance estimation, such as the more accurate power value, and thereby diagnose which of the two measured values can better represent the actual performance of vessels. In this work, the new proposed indicator is the error between the reported values from operational data and the expected values based on baseline models and relevant sailing conditions, which is furthered explained in the next section.

#### 3.2.2. Process to Develop New Indicators

The specific procedure of calculating the new indicator is given in Figure 8. For each data sample in an operation, the expected vessel resistance and the expected power are first calculated through Equation (Equation 2) and Equation (Equation 1), respectively. The Trinity model is calibrated accounting for the reported weather conditions, in terms of wind speed, wave height, and currents, and the assumed hull deterioration [24]. Note that it is necessary to assume such hull deterioration, otherwise the performance loss of the vessel will be always determined by the reported power values. The assumed deterioration is statistically estimated based on the trend of hull performance in the previous reports, taking into account possible noise. Thus, the expected speed–power–RPM curves are obtained for each sample in the operational dataset. All errors of speed, RPM, and power should be evaluated, as all these errors are used for the final source identification. Using these curves, a look-up of expected value (speed, power, and RPM) can be conducted through linear interpolation using the corresponding operational measurements. For instance, based on the measured RPM value, the expected speed and power values can be looked up from the Trinity model. Note that one parameter is always assumed to be correct when the expected values of the other two are estimated. In this fashion, the errors of the speed, power, and RPM for each sample in the dataset can be calculated.

For brevity reasons, the power value based on the measured fuel consumption from the fuel flow meter and the engine performance baseline SFOC (Figure 5) is defined as the fuel-consumption-based estimation method, denoted as the FC method. The power that is the direct measurement from torsion meters using the torque and RPM is the torsion-meter-based estimation method, denoted as TM method.

As there are two estimates of hull deterioration based on the different measured value of power, there are two sets of calibrated curves for each record. Therefore, for the calculation of errors of each parameter, there are two corresponding values when using the two different hull deterioration values. Using the proposed procedure as mentioned above (Figure 8), all the parameter errors can be calculated (see the flowchat in Figure 9). Therefore, it will produce twelve (or six pairs of) errors in total, as seen in Table 1. The same pattern of notation for all parameter errors will be used in this research, for instance, ER_RPM_Speed_TM denotes the calculated RPM error with the given value of speed using the performance deterioration from the TM method.

Since it is not known which of the two signals is better, neither stand-alone error (either with power from TM or with power from FC) can be used to diagnose a faulty signal; instead, we adopt the pair of calculated errors for further analysis. The sliding CV [11] method (see Section 3.3) is applied to detect data aberrations from the error pairs. For instance, the two power errors with given RPM (ER_Power_RPM_FC and ER_Power_RPM_TM) are further analyzed to identify pattern changes and come up with conclusive results.

### 3.3. The Method to Identify Aberrations

In our previous research [11], an original sliding method was proposed to identify unstable periods for a single data variation. In this method, historical CV values of performance indicators bounded by a sliding window with a fixed width and steps were used for the identification of unstable periods. In this study, the same concept is used but with a focus on the detection of aberrations based on a pair of new indicators that can be applied to both noon reports and autolog data. The CV values are calculated within each sliding window to trace changes.

#### 3.3.1. Sliding Window

Figure 10 illustrates the sliding window along the timeline of samples with a window width of *n* and a step size. In this example, the width of five samples is used, and every time it moves forward by two samples, so that there is an overlapping of each sliding window. In practice, the determination of the window width and the step size should be conducted by a sensitivity study for each vessel based on the historical variation in operational data. According to our previous study [11], for most of the vessels from the shipping company TORM [18], a width of thirty samples and a step size of one sample have been adequate. In each sliding window, the CV value is calculated, expressed as the ratio of the standard deviation to the mean value, denoting the extent of data variability around the mean. This method is quite straightforward and the application is simple, as shown in Figure 11. The CV values of the two power errors (denoted as CV_error_FC and CV_error_TM) are directly calculated. The discrepancy of two corresponding CVs in absolute format is then compared, which will trigger alarms when it is violated by a given threshold. Sources can be found by analysis, and crew feedback will be signalled in the operational system (see Section 4.3).

This method can be applied to time series data in different sample frequencies for both online and offline pattern changes identification. Since the sliding window has a width of n samples, the length of dataset (*N*) should have at least n samples, N>n, so that the change along the timeline can be seen and the historical data can be used to determine the threshold and window width for each vessel. When a sample is added, a new calculation is conducted with the moving forward of a sliding window. In this way, the shift can be detected, accounting for the effect of the past n samples. In this regard, a time delay equal to the width of the sliding window may be introduced before a source and the feedback have been given due to triggered alarms. This method is generic in application, i.e., independent of the types of operational data sources such as noon reports or autolog data from sensors.

## 4. Case Studies and Results

To assess the effectiveness of the proposed method and the new indicators, the time series data from vessel A are selected, and the new indicators in terms of power errors are used for analysis. A sensitivity study is first carried out to determine the effect of value signs when performing CV calculations.

### 4.1. Sensitivity Study

By nature, the new indicators (Section 3.2) can take negative values when expected values are smaller than measured values. This largely affects the CV in a sliding window and the application of the practical method. Figure 12 shows an example of the absolute power errors with given RPM from both autolog and noon report data, calculated by the FC method and the TM method, respectively. As shown, the power errors are unequally distributed on both sides of the X-axis. An exceptional case would be that the denominator (mean value) of the CVs approaches zero since the original values are equally distributed on both sides of the X-axis. As a consequence, the corresponding CV would approach infinite. As demonstrated in Figure 13, the CVs with signs are visualized, and spikes of CVs are observed due to infinite values. Therefore, before applying the method, a pre-processing of the power errors needs to be performed, which transforms all values into positives so as to rescale the CVs in a reasonable range for the proposed identification method.

When applying the practical method, another sensitivity study is necessary to determine the proper width and the moving step of the sliding windows, as already illustrated in Section 3.3.1. In our previous study [11], an example has been made of the vessels from the shipping company TORM, where a width of thirty samples and a step size of one sample are selected. Therefore, similar work is not repeated in this paper but the same results are adopted for a further case study. Note that, for different ships, the recommendation is to perform such a sensitivity analysis before utilizing the method.

### 4.2. Identified Aberration Periods

The operational data in vessel A from Hapag-Lloyd are used to identify aberrations by the proposed method. The absolute power errors given the RPM in the two ways for autolog data are presented. Figure 14 illustrates the CV variation in the power errors given the RPM for the period between February and July in 2019. Here, the orange curve denotes the new indicator values calculated by the TM method and the blue curve denotes values by the FC method. Since it is expected that both indicators follow the same trend, any remarkable deviation in the values implies a change or shift in the pattern of the values. The discrepancy between the CV for the two methods can be clearly visualized at a proper scale, and the periods violating a set threshold are highlighted. An alarm will be automatically triggered in this situation. Note that the threshold value is based on the observation of historical data variation and industry experience, which may introduce subjective bias. Periods where the difference between the CV is above threshold indicate that one of the values is not so reliable. The isolation of the source of the deviation is explained in the next section, which will facilitate the selection of more accurate power values.

### 4.3. Deviation Source Analysis

Once an aberration period has been identified, the sources for the identified changes can be further analyzed using baselines (Section 3). For this purpose, the aberrant period from 18 April to 20 May in 2019 (see Figure 14) is selected for detailed analysis.

Figure 15 shows the calculated RPM errors in the detected aberration period estimated by the procedure in Section 3.2. The two horizontal lines in the images are the mean values of the indicator in the given period. The TM method has a lower mean error in the RPM value compared to the FC method, which suggests that the reported power value measured by the torsion meter is more aligned to the previously reported power values from the torsion meter in the source report than those from the fuel flow meter. As presented in this figure, the mean error of RPM based on the speed estimated from the FC method is 2.58 rpm, while the one from the TM method is 1.51 rpm. The mean error of the RPM based on the power estimated from the FC method is 1.77 rpm, while the one from the TM method is 0.30 rpm. Hence, further verification is needed of the fuel consumption values in reports in this period.

Figure 16 is the comparison of the last 10 measured samples from the end of the aberration period in vessel A, which further demonstrates the above conclusion. The speed–RPM dimension is used, as the speed–RPM values for each sample are identical for the two ways and are reliable. The black curve is the original speed–RPM baseline from the light propeller curve, while the blue and the red curves denote the calibrated baselines using FC and TM methods for the last sample, respectively. The measured samples fit better to the curve calibrated by the TM method. The overall analysis thus indicates that the power measured by the torsion meter is more reliable in this identified period and the fuel consumption value has a higher discrepancy and is less reliable to use. Note that the source analysis for the identified aberration periods in this paper is only a demonstration case that was manually conducted to find more reliable values in operations. It is case-dependent for individual vessels. In another vessel, it might be the fuel meter that is more reliable or some errors may occur in either the fuel meters or torsion meters. Thus, extensive domain knowledge in shipping is needed to conduct such an analysis, which limits its industry application. Further work is needed to generalize different aberration periods so that the data aberration source can be automatically identified.

## 5. Discussion

The practical method has been tested on both autolog and noon report data from four different commercial vessels belonging to Hapag-Lloyd, covering 14 vessel years. For confidential reasons, only the final results are summarized here. There are 76 aberration periods found in the case of autolog data and 17 for the noon data based on the new indicators and sliding CV method. Deeper analyses were carried out for all detected periods, where 15.3% of all identified aberrations in autolog data were found to have a conclusive source, with 28.6% in the case of noon reports. This implies that, out of the 76 identified aberration periods for autolog data, 9 were found to have a conclusive source of deviation from either the fuel consumption or the torsion meter. In the case of the noon reports, 4 out of the 17 had a conclusive deviation source. The remainder of the aberration periods were inconclusive due to ambiguity and contradictory conclusions from the methods or too short aberration periods. It is considered that the detection of an aberration period is an important finding as these findings help the selection of more accurate operational parameters for a performance estimation.

To put it into practical terms, there are approximately five aberration periods detected per vessel per year, and there is approximately one triggered period per vessel per year with a conclusive deviation source diagnosed by the proposed method. As no or very little research has been conducted in this field by the shipping industry so far, to the best knowledge of the authors, this is believed to be a significant result. The effectiveness of the proposed method to identify such deviations is of great interest as a start. The derived results in this research are advantageous to implement in diagnostic tools, and the results would be beneficial to facilitate vessel operators and crew in real-time operations.

However, limitations exist for this method and the investigation of sources. The Trinity model may have errors and implicit biases, which need to be kept in mind. In the sliding CV method, the threshold value to trigger alarms is manually defined based on experience and historical data variation, which may introduce subjective bias. A sensitivity study is needed for each vessel or for the same type of vessels to determine the width of the sliding window, which requires manual effort. The method adopts the moving average concept, which means that data quality issues, such as spikes, outliers, and missing values, will have considerable influence on its effectiveness. The sliding CV method is better suited to diagnose a shift or jump but cannot easily identify constant offset values. Finally, the effect of data source differences on the accuracy of the method has not been analyzed, and needs to be further studied. For instance, over a long period of vessel sailing, possible deviations of onboard sensors, such as sensor drift, sensor offset, sensor sensitivity, and sensor hysteresis, may bring uncertainties to the investigation of sources of identified aberrations.

## 6. Conclusions

The comparison between the noon reports and autolog data concludes that the two data sources reflect similar vessel operational behavior but with statistical discrepancies. A practical method to identify aberrations in the time series data from ship operations based on new indicators is presented using the sliding window and CV values. This can be applied for various types of operational data, including noon reports and autolog data. Case studies have been conducted using data from commercial vessels that show how this method can be utilized. Source identification analysis has been performed to discover reasons for the triggered changes. The major contributions of this paper are summarized, as shown in the following.

The noon report and autolog data represent similar vessel operational behavior but with statistical discrepancies due to different data acquisition methods and many uncertainties, such as sensor derivations under extreme environments and human errors.New indicators are proposed from the Trinity model of speed, power, and RPM, where weather and hull deterioration during operation is considered.The practical method based on the proposed new indicators can be applied to identify data aberrations.Despite its time-consuming and manual conduction, the practical deviation source analysis method provides an effective way to select more accurate parameters for ship performance estimation in the complex maritime environment, especially with so many sources of uncertainty during operational data recording.

There are still many deviations that cannot be explicitly explained because of ambiguity, variation tendencies, and noise, which is another big challenge. Further research that focuses on these limitations and drawbacks of the practical method is needed. Applying the method on data—both historical and real-time—can reveal consistent aberration periods that earlier would have been overlooked as noise. The awareness of this shift is a valuable asset to vessel operators and crew, who have better access to analyze the cause of the shift. It has also been demonstrated that the method can be used to isolate the source of the shift in selected cases. Keeping in mind the call for responsible and environmentally sustainable shipping practices, such tools may be immensely powerful in improving performance analyses in the near future. Additionally, this diagnostic tool could facilitate data verification organizations in implementing stricter validation rules for submitted data. Such practical methods in this research are the first step towards bringing a clearer view into the nuances of vessel operational data.

## Figures and Tables

**Figure 1 sensors-24-02146-f001:**
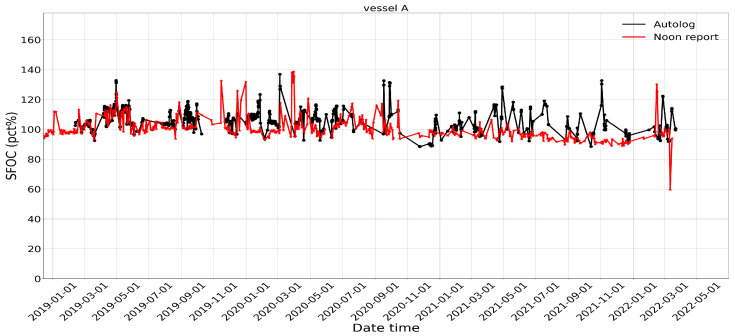
Comparison of the SFOC variation (in percent) of vessel A between noon report and autolog data in a given time period.

**Figure 2 sensors-24-02146-f002:**
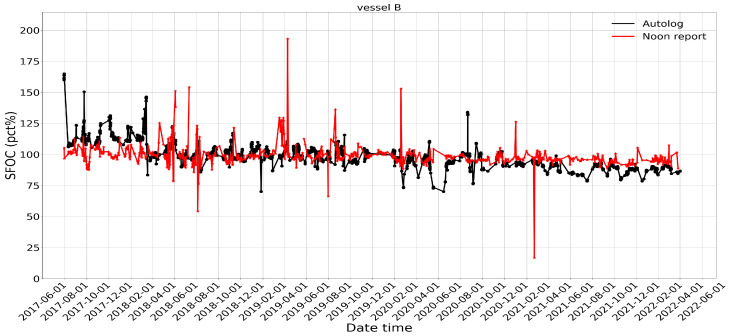
Comparison of the SFOC variation (in percent) of vessel B between noon report and autolog data in a given time period.

**Figure 3 sensors-24-02146-f003:**
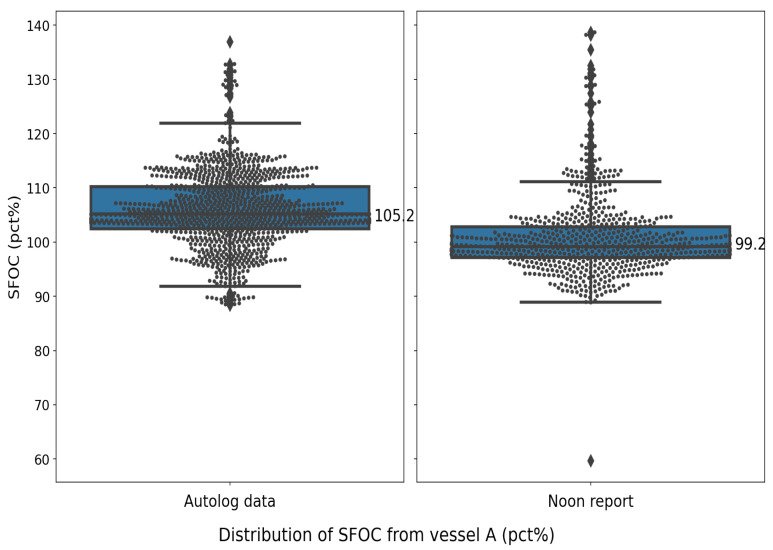
Statistical distribution of vessel A in the given time period.

**Figure 4 sensors-24-02146-f004:**
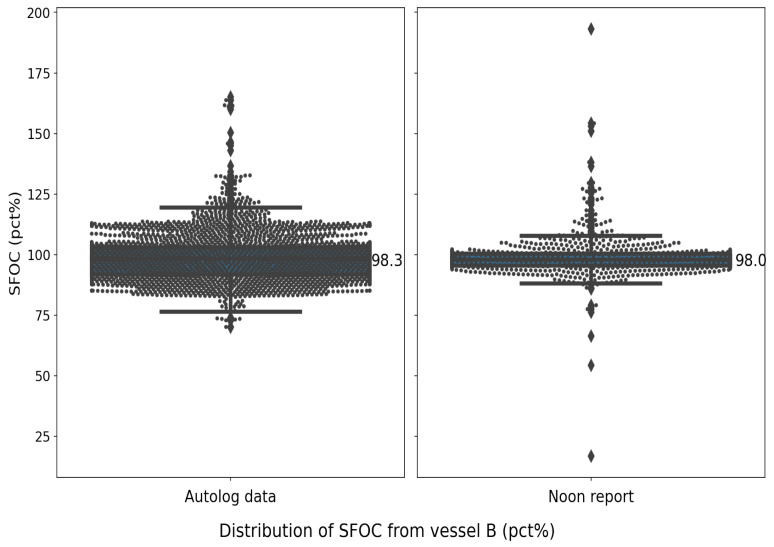
Statistical distribution of vessel B in the given time period.

**Figure 5 sensors-24-02146-f005:**
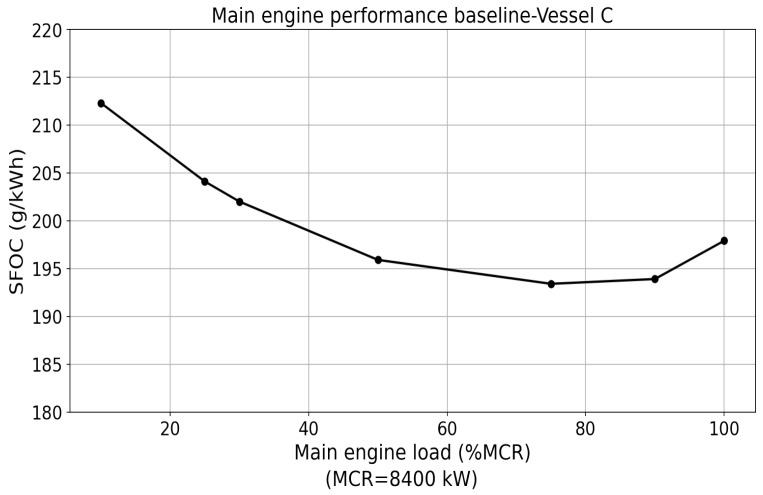
An example of SFOC diagram of the main engine from vessel C obtained from shop test [18].

**Figure 6 sensors-24-02146-f006:**
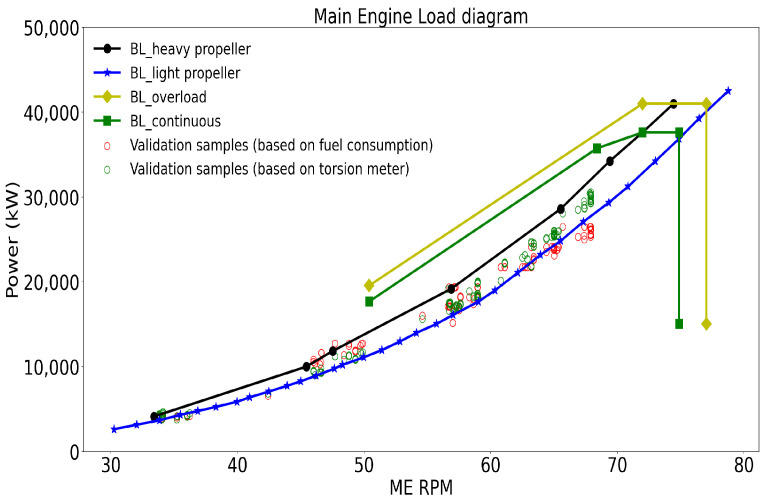
An example of the main engine load diagram from vessel A with validation points from operation [18].

**Figure 7 sensors-24-02146-f007:**
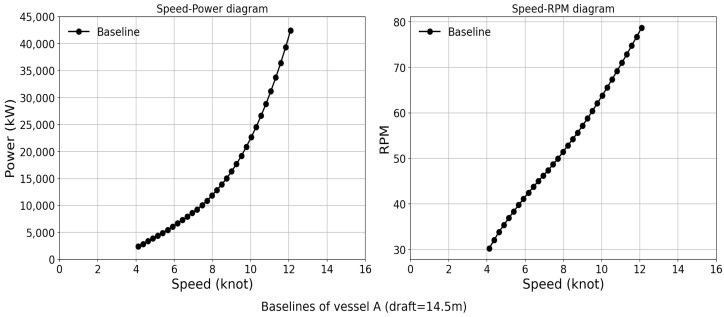
Baselines of the vessel A for speed, power, and RPM.

**Figure 8 sensors-24-02146-f008:**
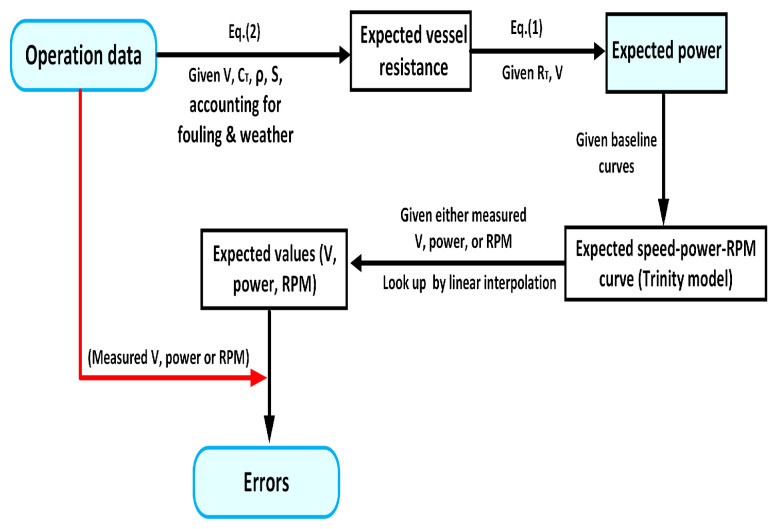
Flowchart of the procedure for the new indicators (parameter errors).

**Figure 9 sensors-24-02146-f009:**
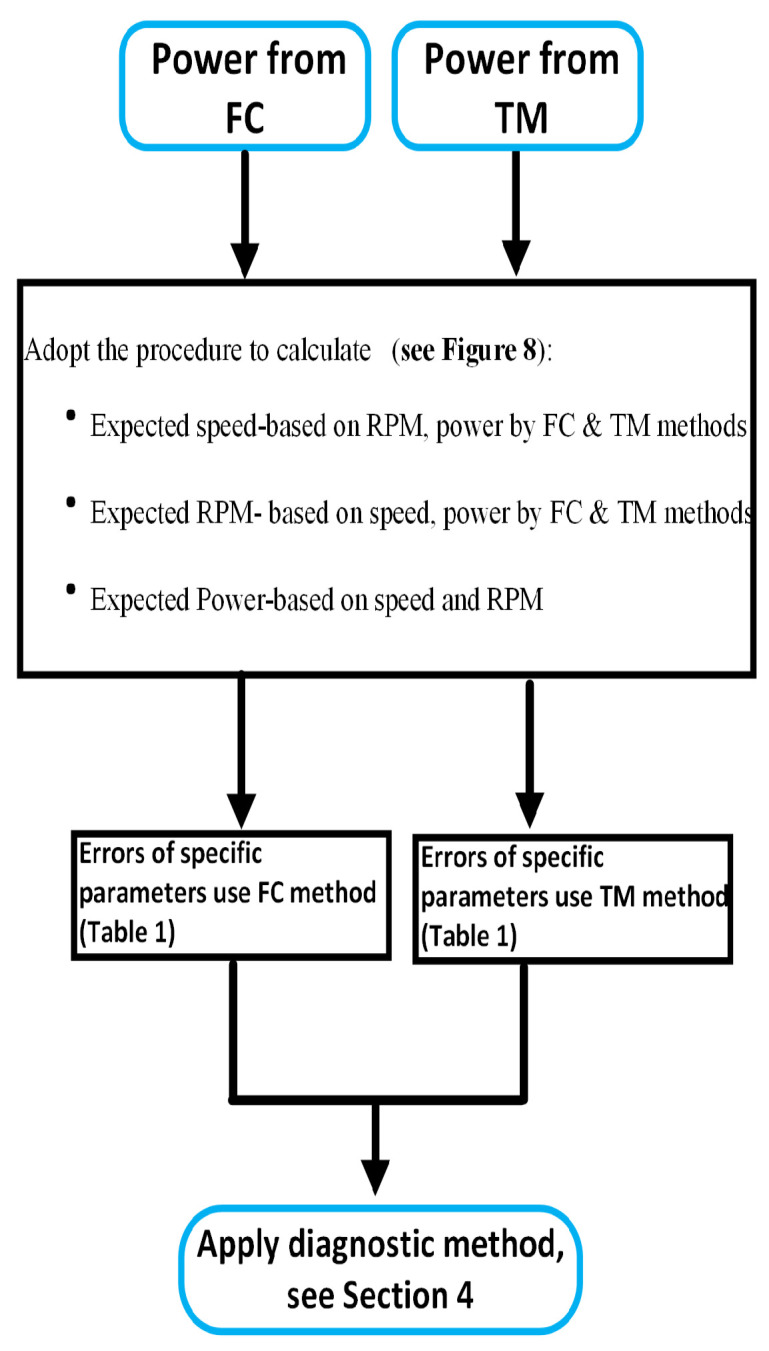
Flowchart of the parameter errors (speed, RPM, and power) calculated from two different power sources (FC and TM) using the new process (Figure 8, Table 1, Section 4).

**Figure 10 sensors-24-02146-f010:**
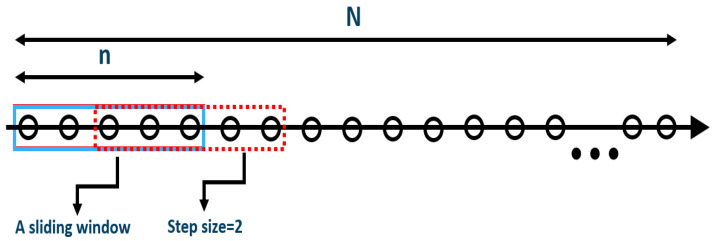
Illustration of the sliding window along the time series data samples.

**Figure 11 sensors-24-02146-f011:**
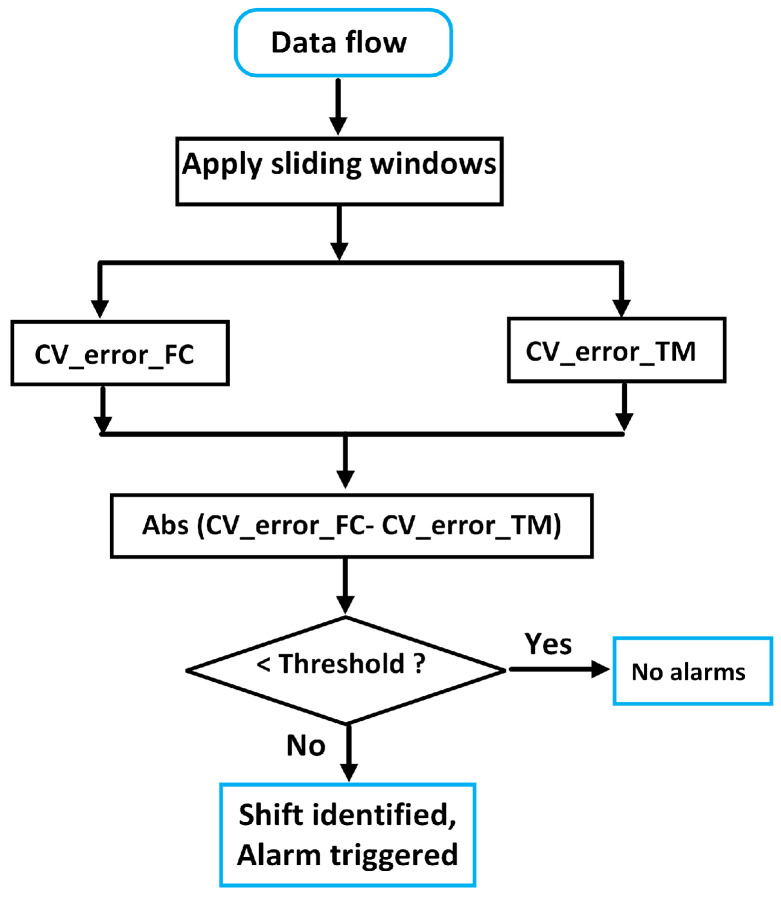
Flowchart of the application of the practical method on the new indicators from two power models.

**Figure 12 sensors-24-02146-f012:**
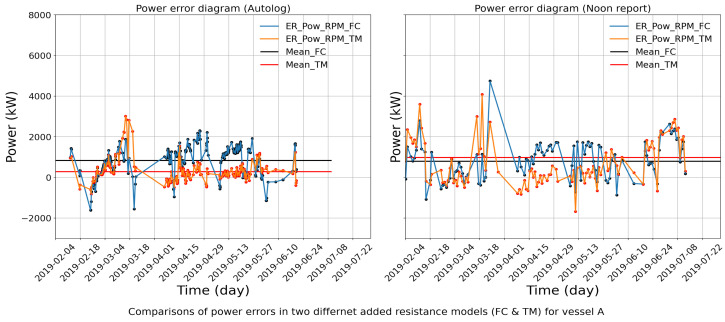
Comparisons of the power errors with signs from FC method and TM method, respectively, for vessel A.

**Figure 13 sensors-24-02146-f013:**
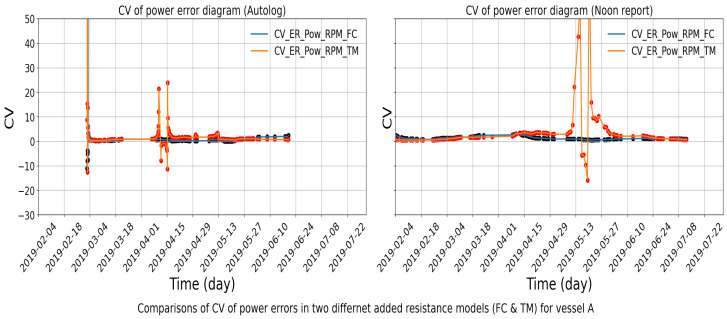
CV of the power errors with signs from FC method and TM method, respectively, for vessel A.

**Figure 14 sensors-24-02146-f014:**
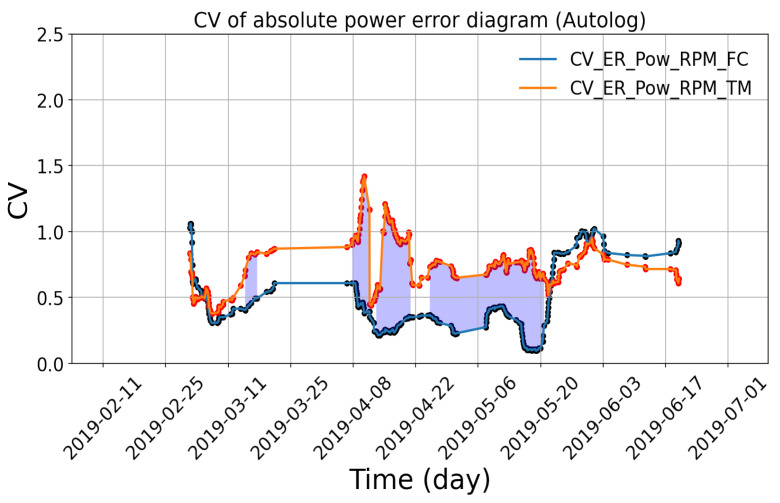
Application of the practical method on power errors (absolute values) for vessel A.

**Figure 15 sensors-24-02146-f015:**
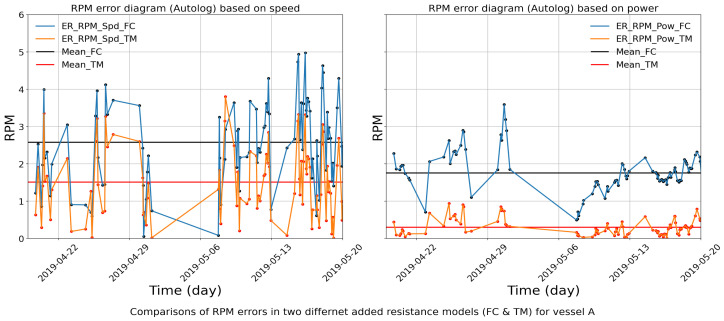
Comparisons of RPM errors from FC method and TM method, respectively, for vessel A.

**Figure 16 sensors-24-02146-f016:**
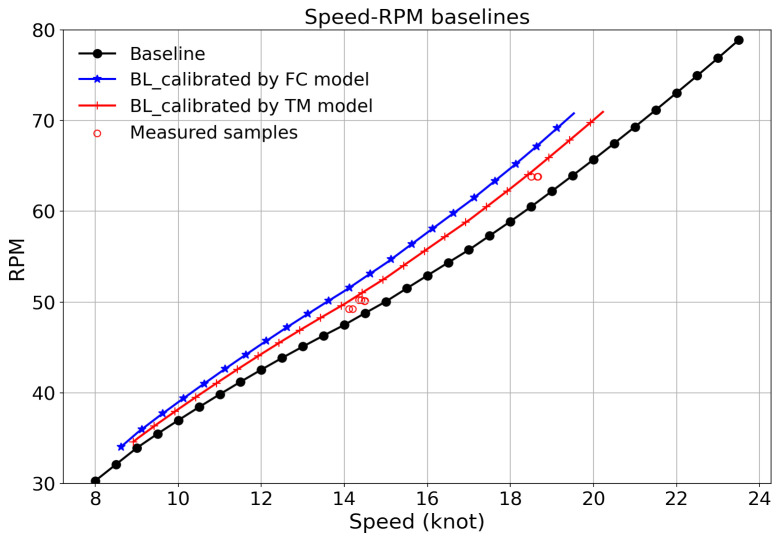
Comparisons between baselines and the last 10 measured samples from the pattern change in vessel A.

**Table 1 sensors-24-02146-t001:** All possible errors based on the new procedure under two different power sources (FC or TM).

Power Source	Error	Given Parameter
	Power error	RPM
	Power error	Speed
FC or TM	RPM error	Power
	RPM error	Speed
	Speed error	RPM
	Speed error	Power

## Data Availability

The noon report data and the autolog data used in this research are unavailable due to confidentiality reasons of the companies.

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
