# Peer review of "Method for Identification of Aberrations in Operational Data of Maritime Vessels and Sources Investigation"

_sensors, 2024, doi:10.3390/s24072146_

Round 1

Reviewer 1 Report

Comments and Suggestions for Authors

This paper presents a method for identifying aberrations in vessel operations and investigating their sources. It compares two types of data sources in vessel operations: noon reports and autolog data. The statistical characteristics of these data sources are compared based on two vessels from the container shipping company Hapag-Lloyd. The article also describes the methodology for identifying data aberrations, including baseline models, traditional indicators for ship performance, new proposed indicators, and the sliding window method. A case study is conducted to apply the proposed method on a commercial vessel. The article concludes with a discussion of the results, method limitations, and draws conclusions.

The completion of this article involved certain data organization and comparisons, presenting some engineering applicability. However, it lacks theoretical depth. Therefore, it is not recommended for acceptance, and the author is advised to rewrite and resubmit.

Comments on the Quality of English Language

none

Author Response

thanks for your possible consideration

Reviewer 2 Report

Comments and Suggestions for Authors

In general the paper provide a solid approach for automatic statistical analysis of operational ship data. The strength of the paper is the practical comparison of manual recording data and data coming from autolog.

Whats unclear is the practical usage of the data. Modern sensor systems are suitable to collect much more data and provide potential for more in deth analysis. Please provide a description of the practical usage of the outcome. 

Author Response

thanks for your possible consideration

Reviewer 3 Report

Comments and Suggestions for Authors

It would be useful to denote type of propeller in methodology. From article, it is to conclude that measurements have been done on fixed pitch propeller, not controllable pitch propeller. 

Page 9,(153) influenced by wind, wave and ....- it would be useful to denote also in condition of depth and currents (as it is in 182). Depth is important because of squat influence. 

Did you include all possible impact to final result? Quality of sensor of RPM instrument, torsion of propeller and small damages of propeller due to exploration in 14 years, trim of hull? This may affect to final result so it is good to notice other parameters in discussion of results. 

Author Response

thanks for your possible consideration

Reviewer 4 Report

Comments and Suggestions for Authors

The paper presents a method for identification of aberrations in vessel operational data coming from manual ship’s noon reports and autolog data from sensors. The experimental design is appropriate and the presented method of sliding windows interesting and with potential of practical applicability. Anyway, my recommendation is to resubmit the article after major revision due to several flaws stated below:

1)      The title is confusing and too broad considering the scope of the research that is restricted to data of vessel's noon report. This should be reflected in the title. The paper as whole should be amended according to MDPI / Sensors paper's template. The Editor should check if the paper was not submitted twice (to Elsevier as well) what is quite unethical.

2)      The abstract should be extended - cover the main numerical results and conclusions as well: What new indicators were proposed. Why the method was effective?

3)      Keywords unnecessary cover the words already included in the title. The authors should use different keywords.

4)      Numbers of the references should start from [1] in the order of their appearances according to MDPI standards.

5)      Lines 21-22 This sentence should be linguistically corrected and its style redesigned. Does a discipline mean an issue?

6)      Line 22 The operational data is better term than vessel operations in general. In most maritime dictionaries vessel operations refer to the various activities and tasks involved in the management and maintenance of a ship. This includes navigation, cargo handling, communication, safety procedures, and maintenance of the vessel, engine, and its equipment. Vessel operations also involve the coordination of crew members and the implementation of standard operating procedures to ensure the safe and efficient operation of the vessel. These operations are critical to the success of any maritime venture, and require skilled personnel with specialized knowledge and training.

7)      Lines 95-96 Statistical analysis should be expanded by formulation of hypothesis and tests to confirm or reject the hypothesis.

8)      Figure 5 & 6: Source of the figures should be added.

9)      Line 157 Physically power equation is P = torque x RPM x 2 x pi. How was the relation for P derived by the authors? What is the meaning of n?

10)   Line 329 Effectiveness of the method should be emphasized in the conclusion by citing numeric statistics from the discussion amended by results of statistical tests as mentioned in 7)

Comments on the Quality of English Language

Some sentences should be amended for clarity.

Author Response

thanks for your possible consideration

Round 2

Reviewer 1 Report

Comments and Suggestions for Authors

The author has made numerous revisions and provided reasonable explanations. The paper has certain practical engineering value, but it is recommended to incorporate some theoretical content for better justification.

Reviewer 4 Report

Comments and Suggestions for Authors

All my comments have been considered in the revised manuscript except the title. In my opinion the title is confusing and too broad considering the scope of the research that is restricted to data of vessel's noon report compared to autolog data. The proper title should be: A Method for Identification of Aberrations in Operational Data of Maritime Vessel and Their Sources Investigation.
